# Developmental genetics of color pattern establishment in cats

Christopher B. Kaelin[1,2,3], Kelly A. McGowan[1,2,3] & Gregory S. Barsh [1,2✉]

Intricate color patterns are a defining aspect of morphological diversity in the Felidae. We applied morphological and single-cell gene expression analysis to fetal skin of domestic cats to identify when, where, and how, during fetal development, felid color patterns are established. Early in development, we identify stripe-like alterations in epidermal thickness preceded by a gene expression pre-pattern. The secreted Wnt inhibitor encoded by *Dickkopf 4* plays a central role in this process, and is mutated in cats with the Ticked pattern type. Our results bring molecular understanding to how the leopard got its spots, suggest that similar mechanisms underlie periodic color pattern and periodic hair follicle spacing, and identify targets for diverse pattern variation in other mammals.

[1] HudsonAlpha Institute for Biotechnology, Huntsville, AL, USA. [2] Department of Genetics, Stanford University School of Medicine, Stanford, CA, USA. [3] These authors contributed equally: Christopher B. Kaelin and Kelly A. McGowan. ✉email: gbarsh@hudsonalpha.org

Understanding the basis of the animal color pattern is a question of longstanding interest for developmental and evolutionary biology. In mammals, markings such as cheetah spots and tiger stripes helped motivate theoretical models, such as the Turing reaction−diffusion mechanism, that have the potential to explain how periodic and stable differences in gene expression and form might arise from a uniform field of identical cells[1–4]. Reaction−diffusion and other mechanisms to account for periodic morphological structures have been implicated in diverse developmental processes in laboratory animals[5–12], but much less is known about mammalian color patterns, largely because the most prominent examples occur in natural populations of wild equids and felids that are not suitable for genetic or experimental investigation.

In fish, color patterns involve direct interactions between pigment cells that are often dynamic, allowing additional pattern elements to appear during growth or regeneration[13–15]. By contrast, in mammalian skin and hair, melanocytes are uniformly distributed during development, and the amount and type of melanin produced are controlled later by paracrine signaling molecules within individual hair follicles[16–18]. Additionally, pattern element identity of an individual hair follicle, e.g., as giving rise to light- or dark-colored hair, is maintained throughout hair cycling and cell division, so that individual spots or stripes of hair apparent at birth enlarge proportionally during postnatal growth. Thus, periodic mammalian color patterns may be conceptualized as arising from a three-stage process: (1) establishment of pattern element identity during fetal development; (2) implementation of pattern morphology by paracrine signaling molecules produced within individual hair follicles; and (3) maintenance of pattern element identity during hair cycling and organismal growth[17,19].

Domestic cats are a useful model to study color patterns due to their accessibility, the genetic and genomic infrastructure, the opportunity for genomic and histological studies of tissue samples, and the diversity of pattern types[19–21]. The archetypal tabby pattern—regularly spaced dark markings on an otherwise light background—varies considerably in both form and color, and many of those varieties are similar to some wild felid species. In previous studies of domestic cats, we showed that Endothelin 3 is expressed at the base of hair follicles in tabby markings, causes a darkening of the overlying hairs by increasing the production of black−brown eumelanin relative to red−yellow pheomelanin, and therefore plays a key role in the implementation of tabby pattern[17]. However, tabby markings are apparent in developing hair follicles[17], indicating that the establishment of the color patterns must occur at or before hair follicle development.

Here, we apply single-cell gene expression analysis to fetal cat skin to investigate the developmental, molecular, and genomic basis of pattern establishment. We uncover an aspect of epidermal development that signifies the establishment of pattern element identity and characterize signaling molecules and pathways associated with pattern establishment. We show that one of those signaling molecules encoded by Dickkopf 4 (Dkk4) underlies a naturally occurring mutation that affects tabby patterns and plays a key role in the patterning process. Our work provides fundamental insight into the mechanisms of color pattern establishment as well as a platform for exploring the biology of periodic patterns more broadly.

## Results

### Patterns of epidermal morphology in developing cat skin.
Trap-neuter-release programs have become a popular means of controlling feral cat overpopulation[22]. During the breeding season, approximately half of all female feral cats are pregnant, and tissue from non-viable embryos can be recovered during spaying without compromising animal health or interfering with efforts to control feral cat overpopulation. We collected more than two hundred prenatal litters from feral cat, spay-neuter clinics across a range of developmental stages classified according to previous work on cats[23] and laboratory mice[24] (Supplementary Table 1).

Histochemical and morphometric analysis revealed an aspect of epidermal development that has not been previously described. At stage 13 (analogous to mouse embryonic day 11, E11), fetal skin is comprised of a uniform monolayer of epithelial cells that covers a pauci-cellular dermis (Fig. 1a). Approximately 16 days later at stage 16 (analogous to mouse E15), before epidermal differentiation and hair follicle morphogenesis, we noticed that the epidermis is organized into alternating regions that are either "thick" or "thin" (Fig. 1a, stage 16). Characterization of keratin expression and cell proliferation indicates that the thick and thin regions are fundamentally different from epidermal stratification that normally occurs later in development (Fig. 1b, c and Supplementary Table 2).

By stage 22 (analogous to postnatal day 4–6 in laboratory mice), well-developed hair follicles are present that can be categorized according to the type of melanin produced (Fig. 1a), and that gives rise to the tabby pattern: dark markings contain mostly eumelanin, while the light areas contain mostly pheomelanin.

To investigate if epidermal thickening at stage 16 might be related to tabby patterns that appear later, we made use of natural genetic variation in Transmembrane aminopeptidase Q, (Taqpep). We showed previously that loss-of-function mutations in Taqpep cause the ancestral pattern of dark narrow stripes ($Ta^M/-$, Mackerel) to expand into less well-organized large whorls ($Ta^b/Ta^b$, Blotched) (Fig. 1d)[17]. $Ta^b$ alleles are common in most feral cat populations[25], and we asked if the topology of stage 16 epidermal thickening is influenced by Taqpep genotype. Two-dimensional maps assembled from 100 μM serial sections of embryos of different genotypes reveal that thick epidermal regions from $Ta^M/Ta^M$ embryos are organized into vertically oriented columns (black bars, Fig. 1d) separated by larger thin epidermal regions (yellow bars, Fig. 1d). By contrast, in $Ta^b/Ta^b$ embryos, the thick epidermal regions in the flank are broadened. A similar observation applies to epidermal topology and tabby patterns in the dorsal neck (Fig. 1d and Supplementary Fig. 1). Thus, embryonic epidermal topologies resemble tabby patterns in adult animals of the corresponding genotype, providing a morphologic signature of color pattern establishment before melanocytes enter the epidermis and before the development of hair follicles. As described below, an associated molecular signature further refines the process to a 4-day window that spans stages 15 and 16, and delineates developmental substages (Supplementary Table 1) that track color pattern establishment more precisely.

### Single-cell gene expression analysis of developing fetal skin.
We hypothesized that the alternating thick and thin regions in fetal epidermis might arise from an earlier molecular pre-pattern. To explore this idea, we dissociated fetal cat skin, enriched for basal keratinocytes, and generated single-cell 3′ RNAseq (scRNAseq, 10x Genomics) libraries.

Among libraries that we sequenced and analyzed, three were from developmental stages (15a, 15b, 16a, Supplementary Table 1) occurring prior to or during the appearance of epidermal thickening. Uniform Manifold Approximation and Projection (UMAP) dimensionality reduction[26] delineates different fetal skin cell populations as distinct cell groups (Fig. 2a and Supplementary Fig. 2), although basal keratinocytes (after enrichment, "Methods"), identified by expression[27] of Krt5, Tp63, and

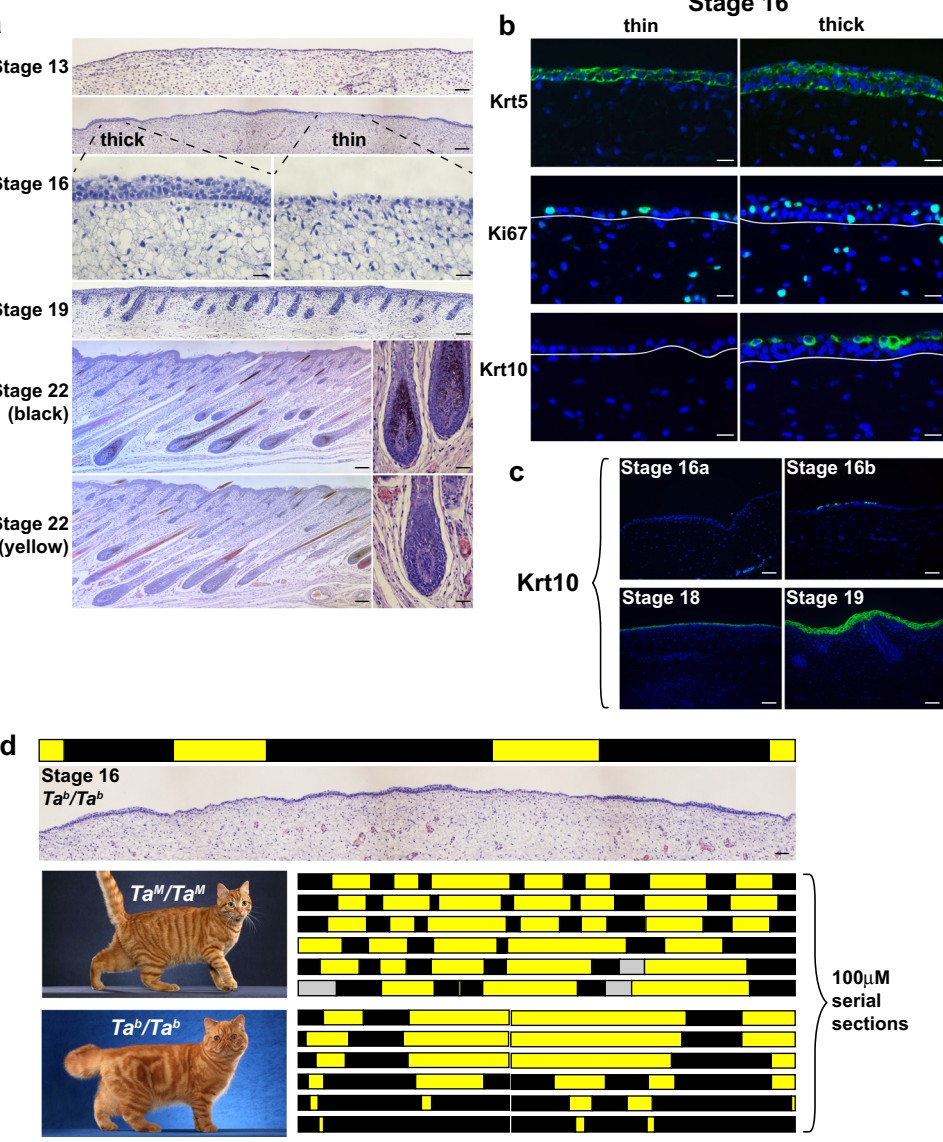

**Fig. 1 Patterns of epidermal thickening in fetal cat skin. a** Cat skin histology at different developmental stages (stage 16, bottom panels—high power fields of the thick and thin epidermis; stage 22, right panels—high power fields of black and yellow follicles). **b** Immunofluorescence for Krt5, Ki67, and Krt10 (green) in stage 16 skin sections (DAPI, blue; white lines mark dermal-epidermal junction). **c** Krt10 immunofluorescence (green) on cat skin at different developmental stages (DAPI, blue). All micrographs (**a**) and immunofluorescence (**b**, **c**) observations were made independently on three or more embryos with similar results. **d** Topological maps based on skin histology (thin, yellow; thick, black) from $Ta^M/Ta^M$ and $Ta^b/Ta^b$ stage 16 embryos mimic pigmentation patterns observed in adult animals (left panels; gray bar—no data). Maps were developed independently on three embryos of each genotype with similar results. Representative images of adult color pattern in $Ta^M/Ta^M$ and $Ta^b/Ta^b$ animals (from Helmi Flick, with permission) are shown on the left. Scale bars: **a** stage 13, 50 μM; stage 16 (top), Stage 19, stage 22 (left) 100 μM; stage 16 (bottom), stage 22 (right) 25 μM; **b** 25 μM; **c**, **d** 50 μM.

$Kremen2$[28], are a major cell type at all three stages (Supplementary Fig. 2 and Supplementary Table 4).

We used unsupervised $k$-means clustering to distinguish different cell types at stage 16a (Fig. 2a, Supplementary Table 3, and Supplementary Fig. 2). At $k = 7$, those clusters correspond to myoblasts, neural crest, dendritic cells, endothelium, dermal fibroblasts, and two types of keratinocytes, a predominant basal population distinguished by high levels of $Krt5$ expression, and a smaller population expressing non-basal markers, including *Keratinocyte Differentiation Associated Protein*, *Rhov*, and *Beta-defensin 1*. At $k = 8$, vascular endothelium can be distinguished from lymphatic endothelium.

At $k = 9$, the basal keratinocytes cluster into two subpopulations based on differential expression of 277 genes (FDR < 0.05, expression >0.25 transcripts/cell), including the secreted

inhibitors of Wnt signaling encoded by *Dickkopf 4* ($Dkk4$, $p = 6.50\mathrm{e}{-46}$, negative binomial exact test) and *Wingless Inhibitory Factor 1* ($Wif1$, $p = 5.81\mathrm{e}{-50}$, negative binomial exact test) (Supplementary Data 1). $Dkk4$ and $Wif1$ exhibit a similar extent of differential expression, 18-fold, and 28-fold, respectively, but in the cells in which those genes are upregulated, the absolute levels of $Dkk4$ are much higher than that of $Wif1$ (36 transcripts per cell compared to 2.2 transcripts per cell, Supplementary Data 1). At $k = 10$, the different UMAP clusters for epithelial cells resolve into distinct populations (Supplementary Fig. 2), but the $Dkk4$-positive (blue) and $Dkk4$-negative cells (green) are not further subdivided.

Unsupervised clustering did not resolve different populations of basal keratinocytes at earlier stages, but a supervised approach based on differential expression of $Dkk4$ (upper and middle

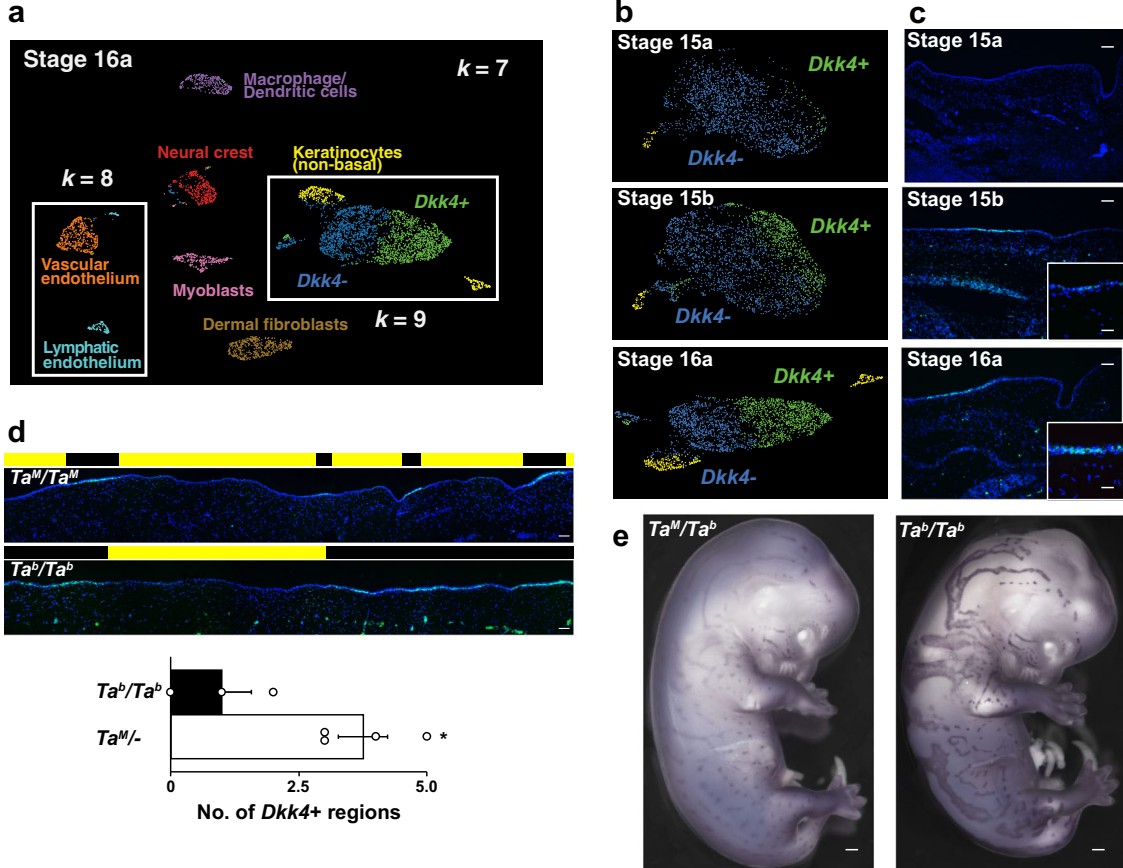

**Fig. 2 Pattern of gene expression in fetal cat skin determined by single-cell RNA sequencing and in situ hybridization. a** Skin cells at stage 16a grouped by their patterns of gene expression (UMAP visualization) with populations colored according to $k$-means clustering from $k = 7$ to $k = 9$ as described in the text. Keratinocytes cluster as non-basal (yellow) and basal populations at $k = 7$, with the latter split into subpopulations distinguished by high (green) and low (blue) *Dkk4* expression at $k = 9$. Subpopulations of endothelium appear at $k = 8$ (orange, cyan). **b** Supervised (top and middle panels; stage 15a and 15b) and $k$-means (bottom panel, stage 16a) clustering at sequential developmental stages delineate non-basal (yellow) and two basal keratinocyte populations, *Dkk4*-positive (green) and *Dkk4*-negative (blue). **c** *Dkk4* expression (green) in fetal cat skin at corresponding stages (DAPI, blue; high power image, inset). In situ hybridizations were carried out independently on three or more embryos at each developmental stage with similar results. **d** *Dkk4* expression (green) in sections of stage 16 $Ta^M/Ta^M$ and $Ta^b/Ta^b$ embryonic cat skin (DAPI, blue). Black and yellow blocks mark thick and thin regions, respectively. The number of *Dkk4*-positive regions (mean $+/-$ sem) in 3.6 mm of Stage 16 $Ta^M/-$ and $Ta^b/Ta^b$ embryonic cat skin from sections ($n = 3$–4 regions from at least two animals of each genotype; two-tail t-test, $^*P = 0.019$). **e** *Dkk4* expression (purple) in stage 16 $Ta^M/Ta^b$ and $Ta^b/Ta^b$ cat embryos. Whole-mount in situ hybridization was carried out on two $Ta^M/Ta^b$ and $Ta^b/Ta^b$ embryo sibs; three additional $Ta^M/-$ embryos showed similar results. Scale bars: **c** 50 μM; **c** inset, 25 μM; **d** 50 μM; **e** 1 mm.

panels, Fig. 2b, "Methods") indicates that the same population apparent at stage 16a can also be recognized at stage 15a and 15b. As with stage 16a, the absolute levels of *Dkk4* expression are the highest of the differentially expressed genes, 22.3 and 99.2 transcripts per cell at stages 15a and 15b, respectively (Supplementary Data 1).

These observations on *Dkk4* in the scRNAseq data were confirmed and extended by in situ hybridization to fetal skin sections, and reveal that *Dkk4* is expressed in alternating subsets of basal keratinocytes at stage 15b, eventually marking the thick epidermal regions that are histologically apparent by stage 16a (Fig. 2c). Between stages 16a and 17, the average size of the *Dkk4* expression domain becomes gradually smaller until it is only apparent in epidermal cells comprising the developing hair germ (Supplementary Fig. 3a, b).

As with the topology of epidermal thickening, we assessed the effect of *Taqpep* genotype on the pattern of *Dkk4* expression at stage 16 and observed a similar outcome: regions of *Dkk4* expression are fewer and broadened in $Ta^b/Ta^b$ embryos compared to $Ta^M/Ta^M$ embryos (Fig. 2d). Qualitatively, the

effect is strikingly apparent from whole-mount in situ hybridization (Fig. 2e and Supplementary Fig. 3c). These embryonic expression differences foreshadow the difference in adult color patterns between $Ta^M/-$ and $Ta^b/Ta^b$ animals. Thus, the expression of *Dkk4* represents a dynamic molecular pre-pattern in developing skin that precedes and is coupled to the topology of epidermal thickening.

**Dynamic changes in Wnt signaling during color pattern establishment.** Additional insight into the developmental mechanisms that underlie the differentiation of basal keratinocytes into *Dkk4*-positive *and* *Dkk4*-negative cells are apparent from patterns of gene expression that distinguish the two populations. Considering stages 15a, 15b, and 16a together, hierarchical clustering of 508 differentially expressed genes (FDR $q$-value < 0.05) highlights groups of genes according to whether they are highly upregulated (cluster A), moderately upregulated (cluster B), or downregulated (cluster C) in *Dkk4*-positive compared to *Dkk4*-negative cells (Fig. 3a). Among the 287 genes that are highly or moderately upregulated, gene ontology enrichment analysis[29]

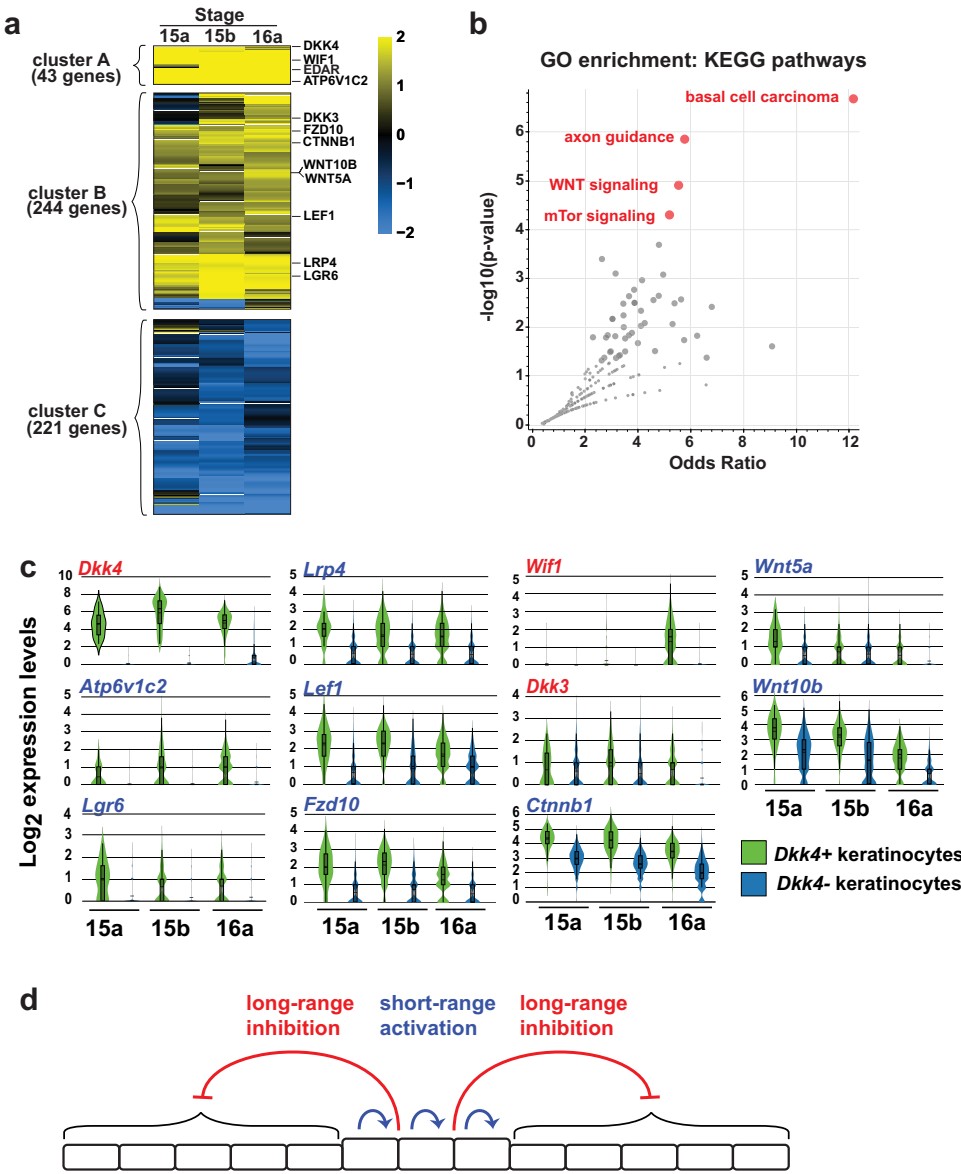

**Fig. 3 Patterns of differential gene expression and gene ontology analysis in basal keratinocyte subpopulations. a** Heatmap of gene expression difference (log₂ fold-change) in *Dkk4*-positive compared to *Dkk4*-negative keratinocytes. **b** Ontology analysis for genes that are highly (cluster A) or moderately (cluster B) upregulated identifies four KEGG pathways that are significantly enriched in *Dkk4*-positive cells. **c** Expression levels for Wnt pathway genes depicted as violin plots. Embedded box and whisker plots depict population maxima and minima (whiskers), first and third quartile boundaries (boxes), mean (solid lines), and median (broken lines). Approach to the analysis of differential expression (**a**) and ontology (**b**) and assessment of individual gene expression levels (**c**) are described in the text. **d** A reaction−diffusion model for color pattern establishment in the basal epidermis where Wnt pathway components participate in both short-range activation and long-range inhibition.

pathways involved in basal cell carcinoma, axon guidance, Wnt signaling, and mTor signaling are the most significant (Fig. 3b). An alternative analysis of differential gene expression with less stringent filtering criteria (expression level change > two-fold) identifies 928, 761, and 606 genes at stages 15a, 15b, and 16a, respectively, of which 121 upregulated and 63 downregulated genes are in common across all stages, and also highlights components of Wnt signaling (Supplementary Fig. 4a–c).

Expression levels for individual Wnt pathway genes at each stage are depicted in Fig. 3c[30,31] and reveal that both inhibitors (*Dkk4*, *Wif1*, *Dkk3*) and activators (*Lrp4*, *Wnt5a*, *Atp6v1c2*, *Lef1*, *Wnt10b*, *Lgr6*, *Fzd10*, and *Ctnnb1*) of Wnt signaling are upregulated in *Dkk4*-positive compared to *Dkk4*-negative cells. Notably, however, the activators encode proteins that are either not secreted (*Lrp4*, *Atp6v1c2*, *Lef1*, *Lgr6*, *Fzd10*, and *Ctnnb1*) or

have a very short radius of action (*Wnt5a*, *Wnt10b*) while the inhibitors encode secreted proteins with a wider radius of action[12]. These observations suggest a reaction-diffusion model for the establishment of color patterns in which *Wnt10b* and *Wnt5a* serve as short-range activators, and *Dkk4*, *Dkk3*, and *Wif1* serve as long-range inhibitors (Fig. 3d).

Direct evidence of Wnt activation in *Dkk4*-positive cells is apparent from comparing the patterns of Ctnnb1 (β-catenin) expression and cellular localization in thick (*Dkk4*-positive) and thin (*Dkk4*-negative) regions of fetal epidermis at stage 16a (Fig. 4a). In thin regions of the epidermis, Ctnnb1 immunostaining is confined mostly to the cell membrane, but in thick regions, ~80% of the basal keratinocyte nuclei exhibit cytoplasmic and/or nuclear β-catenin immunostaining. In addition, expression of *Edar*, a direct target of Wnt signaling during skin development in

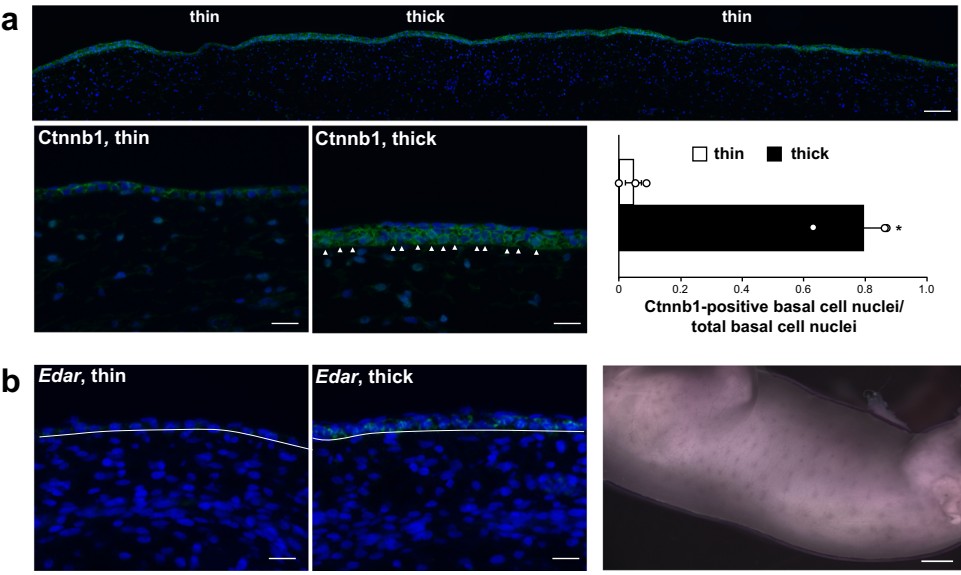

**Fig. 4 Ctnnb1 immunostaining and _Edar_ expression in embryonic cat skin. a** Ctnnb1 immunofluorescence staining (green) in thin and thick epidermal domains from sections of Stage 16a embryonic cat skin (DAPI, blue). White arrow heads mark Ctnnb1-positive basal cell cytoplasm and/or nuclei. The bar graph shows the fraction of Ctnnb1-positive basal cells (of a total number of basal cells; mean $+/-$ sem) in thin and thick domains ($n = 3$ Stage 16a embryos, basal cells from three thick and three thin regions from each embryo were evaluated; two-tailed $t$-test, $*P = 0.006$). **b** _Edar_ expression (green) in thin and thick epidermis from sections of a stage 16a embryo (similar results from Stage 16a ($n = 2$) and Stage 16b ($n = 3$) embryos; DAPI, blue) and _Edar_ expression (purple) in a Stage 16b embryo ($n = 1$). Scale bars: **a** (top), 100 μM; **a** (lower left and middle), 25 μM; **b** (left, middle), 25 μM; **b** (right), 1.5 mm.

mice[32], is increased in sections of thick compared to thin areas of stage 16a epidermis (Fig. 4b) and exhibits a pattern of expression in whole-mount staining that is similar to that of _Dkk4_ (Fig. 2e).

In laboratory mice, a Wnt-Dkk reaction−diffusion system has been suggested previously to underlie periodic hair follicle spacing, based on the patterns of gene expression and gain-of-function transgenic experiments[12,33]. In fetal cat skin, epidermal expression domains of _Dkk4_ are much broader and appear earlier than those associated with hair follicle placode spacing (Supplementary Fig. 3a, b), although by stage 17, expression of _Dkk4_ in cats is similar to the expression of _Dkk4_ in mice.

We compared genes that were differentially expressed in basal keratinocyte subpopulations during color pattern establishment in cats with those that are differentially expressed between hair follicle placodes and interfollicular epidermis in mice, based on a previously established 262 gene signature for primary hair follicle placodes[30]. Of the 121 genes that we identified as a signature for color pattern establishment in _Dkk4_-positive basal keratinocytes, 26 (21.5%) were shared with the primary hair follicle placode signature (Supplementary Data 1). Thus, the gene expression pre-pattern for color pattern establishment occurs prior to and foreshadows an overlapping pattern in hair follicle placodes.

**A _Dkk4_ mutation in domestic cats.** The mackerel stripe pattern represents the ancestral state; in addition to the blotched pattern caused by _Taqpep_ mutations (Fig. 1d), several additional pattern types are recognized in domestic cats for which the genetic basis is uncertain. For example, periodic dark spots as seen in the Egyptian Mau or Ocicat breeds (Fig. 5a) are only observed in $Ta^M/-$ animals[19,34], but the genetic basis of Mackerel vs. Spotting is not known. Another locus, _Ticked_, named for its ability to prevent dark tabby markings and thereby showcase hair banding patterns across the entire body surface, has been selected for in breeds with a uniform appearance such as the Abyssinian, Burmese, or Singapura (Fig. 5a)[35]. However, _Ticked_ is also thought to be responsible for the so-called "servaline" pattern of spotted Savannah cats (Fig. 6a), in which large dark spots are reduced in

size and increased in number. _Ticked_ was originally thought to be part of an allelic series that included $Ta^M$ and $Ta^b$ [36], but was subsequently mapped to an independent locus on chrB1, and recognized as a semidominant derivative allele, $Ti^A$, which obscures tabby markings except on the legs and the tail when heterozygous, and eliminates tabby markings when homozygous[19,37].

_Ticked_ maps to a region of low recombination close to the centromere, and has therefore been difficult to identify by linkage analysis. Furthermore, we observed that some cats thought to be homozygous for _Ticked_, including Cinnamon, the source of the feline reference sequence, carried two haplotypes across the linkage region, suggesting the existence of multiple $Ti^A$ mutations. We hypothesized that one or more of the genes that was differentially expressed between basal keratinocyte subpopulations might represent _Ticked_, and asked whether any differentially expressed gene satisfied two additional criteria: (1) a genetic map position close to or coincident with _Ticked_; and (2) carries a deleterious variant found at high frequency in breeds with a Ticked phenotype. Of the 602 differentially expressed genes at stage 16a (FDR $q$-value < 0.05, Supplementary Data 1), four are located in a relatively small domain, ~230 kb, that overlaps the _Ticked_ linkage interval (Fig. 5b). One of these genes, _Dkk4_, exhibits a pattern of variation and association consistent with genetic identity as _Ticked_.

We surveyed the 99 Lives collection of domestic cat whole-genome sequence data[38], and identified two nonsynonymous variants in _Dkk4_, p.Ala18Val (felCat9 chrB1:42620835c>t) and p.Cys63Tyr (felCat9 chrB1:42621481g>a), at highly conserved positions (Supplementary Fig. 5 and Supplementary Table 5) that were present only in breeds with obscured tabby markings (Abyssinian, $n = 4$; Burmese, $n = 5$; Siamese $n = 1$). The p.Ala18Val variant is predicted to impair signal peptide cleavage, which normally occurs between residues 18 and 19, and the p.Cys63Tyr variant is predicted to disrupt disulfide bonding in a cysteine knot structure referred to as CRD1[39] (Fig. 5c and Supplementary Fig. 5). These are the only coding alterations predicted to be deleterious in any of the four genes that are

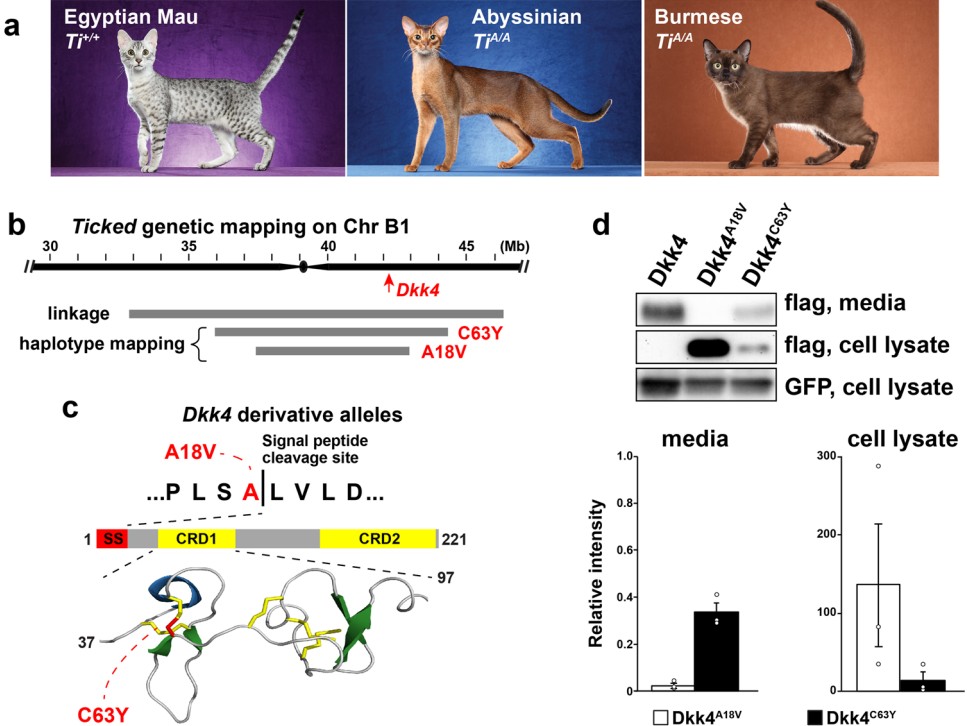

**Fig. 5 Dkk4 mutations in Ticked cats. a** Cat breeds selected for Spotted (Egyptian Mau) or for Ticked (Abyssinian and Burmese) color patterns, along with their inferred *Ticked* genotypes. Images from Helmi Flick, with permission. **b** *Ticked* genetic interval delineated by linkage[19] or shared haplotypes among cats with *Dkk4* mutations. Coordinates are based on the felCat9 assembly. **c** *Dkk4* coding mutations (red) at the signal peptide cleavage site (p.Ala18Val) or a cysteine residue (p.Cys63Tyr) involved in disulfide bridge and cysteine knot formation within cysteine-rich domain 1 (CRD1). **d** Representative Western blot and quantitation of variant Dkk4 proteins produced by HEK293 cells 48 h after transfection. Dkk4 proteins are detected with an anti-FLAG antibody; co-transfection with a GFP-expressing plasmid and subsequent detection of GFP provides a control for transfection efficiency. Relative intensity on the *y*-axis refers to the ratio of the Dkk4 band to GFP band intensity, normalized to non-mutant Dkk4; thus non-mutant Dkk4 relative intensity = 1.0 for both media and cell lysate. The experiment was repeated three times (mean +/− sem, one-way ANOVA, $F = 515.2$, $P = 1.94 \times 10^{-7}$ for media).

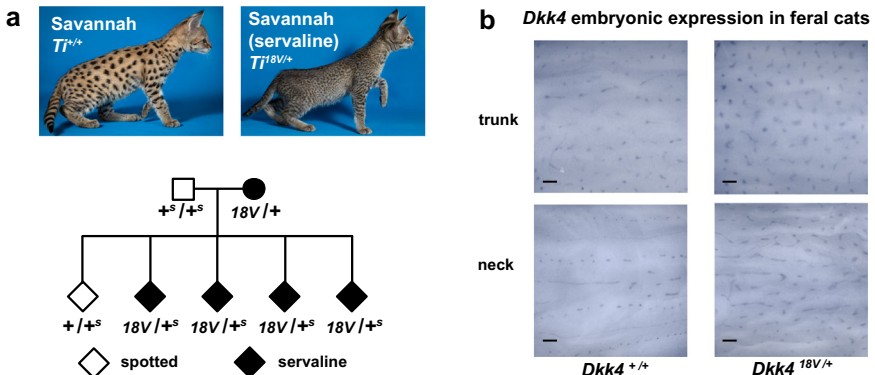

**Fig. 6 Effect of a Dkk4 mutation on color pattern and Dkk4 expression in fetal skin. a** Phenotype of spotted and servaline Savannah cats (from Jamila Avaega, with permission), and cosegregation of those phenotypes with *Dkk4* p.Ala18Val. The domestic cat and Serval non-mutant alleles are represented by + and +$^s$, respectively. **b** Altered *Dkk4* expression (dark markings) in whole-mount preparations from stage 17 fetal skin. Whole-mount expression of *Dkk4* was made on multiple body regions from the trunk and neck of paired +/+ and +/A18V embryos. Scale bar, 300 μM.

differentially expressed in basal keratinocyte subpopulations (Supplementary Table 5) and are therefore strong candidates for *Ticked*.

We evaluated the impact of the p.Ala18Val and p.Cys63Tyr variants on Dkk4 production and secretion by expressing epitope-tagged normal and variant proteins in heterologous cells, and compared the amount of protein retained in cells and secreted into media after 48 h. As depicted in Fig. 5d and relative to non-mutant protein almost none of the p.Ala18Val variants is

secreted, and secretion of the p.Cys63Tyr variant is reduced ~60%; reciprocal results are apparent in the extent of Dkk4 intracellular retention.

Association analysis of additional DNA samples across breeds (*n* = 105) and within breeds (*n* = 238) (Table 1) provides further support that variation in *Dkk4* causes *Ticked* (Table 1 and Supplementary Table 6, "Methods"). All breed cats in which *Ticked* is required (Abyssinian, Singapura) carried the p.Ala18Val or p.Cys63Tyr *Dkk4* variants, the vast majority as homozygotes or

**Table 1 *Dkk4* genotypes in Ticked and non-Ticked cats.**

| Breed and phenotype | $Dkk4^{+/+}$ | $Dkk4^{+/-}$ | $Dkk4^{-/-}$ | Total |
|---|---|---|---|---|
| Abyssinian (Ticked) | 0 | 1 | 36 | 37 |
| Singapura (Ticked) | 0 | 2 | 24 | 26 |
| Burmese (Ticked) | 1 | 1 | 11 | 13 |
| Subtotal | 1 | 4 | 71 | 76 |
| Egyptian Mau (non-Ticked) | 8 | 0 | 0 | 8 |
| Ocicat (non-Ticked) | 13 | 0 | 0 | 13 |
| Bengal (non-Ticked) | 10 | 0 | 0 | 10 |
| Subtotal | 31 | 0 | 0 | 31 |
| OSH and OLH Ticked[a] | 0 | 23 | 2 | 25 |
| Non-breed cats Ticked | 0 | 4 | 0 | 4 |
| Subtotal | 0 | 27 | 2 | 29 |
| OSH and OLH non-Ticked[a] | 21 | 0 | 0 | 21 |
| Non-breed cats non-Ticked | 182 | 0 | 0 | 182 |
| Subtotal | 203 | 0 | 0 | 203 |

[a]Oriental short hair and oriental long hair breeds.

compound heterozygotes. By contrast, in breeds in which tabby markings are required (Egyptian Mau, Ocicat, Bengal), none carried a derivative *Dkk4* allele. Lastly, in some breeds (Oriental Longhair, Oriental Shorthair) and in non-breed cats, either Ticked or non-Ticked phenotypes are observed, and are perfectly correlated with the presence or absence of the p.Ala18Val *Dkk4* variant.

Haplotype analysis indicates that the p.Ala18Val and p.Cys63Tyr variants occurred on common and rare haplotypes, respectively, and like the linkage interval, are relatively long, 6.08 and 8.95 Mb, respectively (Fig. 5b and Supplementary Fig. 6a). We identified several small pedigrees where one or both of the *Dkk4* variants are present (Fig. 6a and Supplementary Fig. 6b, c), and observed segregation patterns consistent with the genetic identity of *Dkk4* as *Ticked*. A pedigree of Savannah cats in which the servaline phenotype cosegregates with the p.Ala18Val variant is of particular interest (Fig. 6a): rather than an apparent suppression of tabby markings as in the Abyssinian and Burmese breeds, *Ticked* alters the number and size of tabby markings.

Unlike *Taqpep*, variation for *Ticked* in California feral cat populations is rare (Supplementary Table 6); however, one of our fetal skin samples was heterozygous for the p.Ala18Val variant, enabling us to examine the effect of *Dkk4* on its own expression. As depicted in Fig. 6b, whole-mount in situ hybridization of a *Dkk4* probe to fetal skin of $Ti^{+/+}$ and $Ti^{+/A18V}$ individuals yields patterns that are remarkably similar to the adult Savannah and servaline color markings, respectively. Taken together, these results confirm that the effect of *Ticked* is not to mask dark tabby markings (as it appears to do in the Abyssinian and Burmese breeds) but to affect pattern establishment such that the regions that express *Dkk4* in fetal keratinocytes, and the eventual adult markings, become smaller and more numerous.

## Discussion

Key elements of reaction–diffusion models as applied to biological patterns are the existence of diffusible signaling molecules that interact with one another to achieve short-range activation and long-range inhibition[40]. Our work identifies molecular candidates for this process in the establishment of tabby patterns, suggests that similar mechanisms underlie periodic color patterning and periodic hair follicle spacing, and provides a genomic framework to explore natural selection for diverse pattern types in wild felids.

The development of tabby patterns is different and arguably simpler than developmental systems that involve extensive cell movement or cytoplasmic protrusions, as in zebrafish[13,15,41].

Additionally, the establishment of tabby patterns is completely and remarkably uncoupled from the implementation events that follow days and weeks later. In particular, epidermal expression of *Dkk4* is dynamic, marking areas of fetal skin that give rise to hair follicles that later produce dark pigment throughout successive hair cycles. This implies that *Dkk4*-expressing keratinocytes acquire time-sensitive epigenetic changes that are later incorporated into hair follicles, and that ultimately determine whether the underlying dermal papilla releases paracrine agents that darken or lighten the hair.

The molecular and the morphologic signatures of tabby pattern establishment recede by stage 17 as hair follicle placodes start to form. Nonetheless, the two processes share important features. A Wnt-Dkk reaction–diffusion system has been proposed to underlie hair follicle spacing[12], and similarity between the transcriptional profiles described here and those in hair follicle placodes[30] suggests the two processes could be mediated by similar mechanisms. Previous work on hair follicle spacing involved a gain-of-function perturbation in which overexpression of *Dkk2* in incipient hair follicles caused both a reduced number and a clustered distribution of follicles[12]. Our work on *Dkk4* mutations yields reciprocal results: the p.Ala18Val and p.Cys63Tyr alleles are loss-of-function and, in servaline cats, are associated with both an increased number and reduced size of dark spots.

In domestic cats, the servaline phenotype is one of several examples demonstrating that the effect of *Dkk4* depends on genetic background and genetic interactions. In addition, the different appearance of the Abyssinian and Burmese breeds and the interaction of the *Ticked* (*Dkk4*) and *Tabby* (*Taqpep*) genes[19,42] further support his notion. In the Abyssinian breed, *Ticked* is selected for its ability to draw attention to prominent hair banding patterns. By contrast, in the Burmese breed, individual hairs are not banded due to a loss-of-function *Agouti* allele[43], and *Ticked* has been selected for its ability to suppress tabby markings. Stated differently, *Dkk4* is epistatic to *Taqpep*, such that the distinction between a Blotched and Mackerel phenotype can only be visualized in a non-Ticked background. [17]*Taqpep* encodes an aminopeptidase whose physiologic substrates have not been identified[44] and as shown here (Supplementary Data 1), *Taqpep* is expressed in developing skin[17]. These observations are consistent with a model in which Taqpep-dependent cleavage restricts the range and/or activity of Dkk4, representing a previously unknown mechanism for modifying Wnt signaling.

Variation in *Taqpep* is associated with pattern diversity in the cheetah as well as in the domestic cats[17], and the same may be true for *Dkk4*. In fact, the servaline pattern was first described in 1839 as characteristic of a new species, *Felis servalina*, but was later recognized as a rare morph of the Serval[45]. We examined *Dkk4* predicted amino acid sequences for 29 felid species including the Serval, and identified a number of derived variants (Supplementary Fig. 7). Although none of the derived variants represents an obvious loss-of-function, several are predicted to be moderately deleterious (Supplementary Table 7), and a more comprehensive survey of *Dkk4* variation in Servals could reveal additional mutations that explain the spotting pattern of *Felis servalina*.

More broadly, genetic components of the molecular signature we identified for the color pattern establishment are potential substrates for natural selection and evolution of pattern diversity in wild felids and other mammals[46]. For example, complex patterns such as rosettes in the jaguar or complex spots in the ocelot may represent successive waves of a Wnt-Dkk system, similar to what has been proposed for hair follicle spacing[12]. Although the action of *Dkk4* in domestic cats appears limited to the color

pattern, subtle pleiotropic effects of *Dkk4* inactivation that reduce fitness may not be apparent. If so, regulatory variation in candidate genes identified here may be substrates for diversifying selection during felid evolution.

Our work exemplifies the potential of companion animal studies to reveal basic insight into developmental biology, and builds on a history in model organisms such as laboratory mice, in which coat color variation has been a fruitful platform for studying gene action and interaction. The Ticked, Tabby, and Blotched phenotypes represent a fraction of the pattern diversity that exists among domestic cat breeds and interspecific hybrids, which remain a continued resource to explore how spots and stripes form in nature.

## Methods

**Biological samples**. Otherwise discarded embryonic cat tissues were recovered from incidentally pregnant feral cats at spay-neuter clinics in California after the spaying surgery. Tissues were processed within 18 h of the spaying procedure and embryos were staged based on the crown-rump length and anatomic development (Supplementary Table 1)[23]. Genomic DNA for genetic association studies was collected with buccal swabs (Cyto-Pak, Medical Packaging Corporation) at cat shows with permission of the owner or by mail submission from breeders. All animal work was conducted in accordance with relevant ethical regulations under a protocol approved by the Stanford Administrative Panel on Laboratory Animal Care (protocol APLAC-9722).

Genomic DNA was extracted from buccal swabs or tissues using a QIAamp DNA kit (Qiagen). Genotyping for *Taqpep* p.Trp841X or *Dkk4* p.Ala18Val and p.Cys63Tyr is based on Sanger sequencing of PCR amplicons (Supplementary Table 8).

**Embryonic staging and characterization of epidermal morphology**. Previous work has classified the domestic cat's 63-day gestation into 22 developmental stages[23]. Supplementary Table 1 depicts the criteria we used for staging, including size (crown-rump length) and visible morphological features such as digit formation and eyelid morphogenesis. Additional features described here, including the acquisition of epidermal thickening, immunohistochemistry for Krt10, and expression of *Dkk4* as assessed by in situ hybridization, were used to refine stages 15 and 16 into substages. Stage 15b is distinguished from 15a by the expression of *Dkk4*. (We note that stage 15a scRNAseq identifies a small number of cells that express *Dkk4*, but that are not detectable by in situ hybridization (Fig. 2b, c).) In the skin, stage 16a is distinguished from 15b by the appearance of epidermal thickening (and expression of *Dkk4* in the thickened regions). Stage 16b is distinguished from 16a by the appearance of some Krt10-positive cells in thick regions.

At stages 16a and 16b, thick regions have 3–5 layers of monomorphous, basaloid cells, whereas thin regions have 1–2 cell layers. Several observations indicate that the thick and thin regions are fundamentally different from epidermal stratification that normally occurs later in development: (1) suprabasal keratinocytes in both regions express a keratin, Krt5, that is normally limited to basal keratinocytes[27]; (2) cell proliferation as indicated by the expression of Ki-67 occurs in all epidermal layers rather than being limited to basal keratinocytes; and (3) Krt10, which is normally distributed uniformly in suprabasal keratinocytes at stage 18 and beyond (Fig. 1c), exhibits an unusual pattern of expression in thick regions towards the end of stage 16 (Fig. 1b, c and Supplementary Table 1).

Stage 17 is distinguished from stage 16b by fetal size, eyelid closure, and the initiation of hair follicle development (Supplementary Table 1). In addition, between stages 15b and 17, the size of the *Dkk4*+ expression domain becomes smaller and is eventually restricted to the hair bud (Supplementary Fig. 3). Changes in epidermal morphology and expression of *Dkk4* in stages 15 and 16a are not observed at the corresponding stages in laboratory mice. Morphological differences among these previously unknown aspects of cat epidermal development at stages 15 and 16, and normal hair follicle development and epidermal stratification that occurs later in other mammals (including the cat) are summarized in Supplementary Table 2.

We assessed the effect of *Taqpep* genotype on epidermal topology by morphometric reconstruction from serial sections from the flank (Fig. 1d, three maps from each genotype) and the dorsal neck (Supplementary Fig. 1, one map from each genotype) of $Ta^M/-$ and $Ta^b/Ta^b$ embryos. The orientation of tabby stripes with respect to the body axis differs between the flank and the neck, but in both cases, the patterns of fetal epidermal topology and their response to variation in *Taqpep* genotype correspond to stereotypic postnatal color patterns. Specifically, in fetal skin from the flank, thick epidermal regions are organized into columns that are perpendicular to the body axis (black bars, Fig. 1d); in fetal skin from the dorsal neck, thick epidermal regions lie in narrow, vertical columns that are parallel to the body axis (black bars, Supplementary Fig. 1). In both cases, the thick regions are expanded in $Ta^b/Ta^b$ embryos.

**Histology and immunofluorescence**. Embryonic cat tissues were fixed in 4% paraformaldehyde (Electron Microscopy Sciences). 5 μM paraffin-embedded sections were mounted on Superfrost Plus glass slides (Fisher Scientific, Pittsburgh, PA) and stained with hematoxylin and eosin. Immunofluorescence for Krt5 (Biolegend 905501, 1:15000), Krt10 (Abcam ab76318, 1:15000), Ki67 (Abcam ab15580, 1:15000), and β-catenin (BD Biosciences 610154, 1:5000) was carried out on 5 μM sections after antigen retrieval using 0.01 M citrate buffer, pH 6 in a pressure cooker. Krt5-stained sections were incubated with goat anti-rabbit Alexa488 antisera (Jackson ImmunoResearch Laboratories 111-545-144, 1:400). Krt10 and Ki67 stained sections were incubated with goat anti-rabbit biotinylated antisera (Jackson ImmunoResearch Laboratories 111-065-144, 1:5000), and β-catenin stained sections were incubated with goat anti-mouse biotinylated antisera (Jackson ImmunoResearch Laboratories 111-065-166, 1:1000), followed by Vectastain Elite Avidin-Biotin complex reagent (Vector Labs), and Tyramide-Cy3 amplification reagent (PerkinElmer). Sections were stained with ProLong antifade reagent with DAPI (Invitrogen). Images were captured on a Leica DMRXA2 system with DFC550 camera and LASV4 (v.4.2.0) software. All photomicrographs are representative of at least three animals at each developmental time point or genotype, except for *Dkk4* expression in the *Dkk4* mutant background; only a single p.Ala18Val embryo was recovered, and observations were made on multiple skin fragments from two body regions (Fig. 6b). Precise genotypes are provided in the text and figures, except in instances where $Ta^M/Ta^M$ and $Ta^M/Ta^b$ embryos were used together, which are designated as $Ta^M/-$.

**In situ hybridization**. Digoxigenin-labeled RNA probes were generated from a region spanning exons 1–3 of cat *Dkk4* and exons 4–6 of cat *Edar* using a PCR-based template (cDNA from Stage 16 cat embryonic epidermal cells; CCCTGAGTGTTCTGGTTTT, AATATTGGGGTTGCATCTTCC for *Dkk4* and ATGGCACCAAAGATGAGGAC, GGCTCCTGTACATTCCTTGG for *Edar*) and in vitro transcription (Roche Diagnostics). 10 μM paraffin-embedded sections were deparaffinized with xylenes, followed by antigen retrieval using 0.01 M citrate buffer, pH 6 in a pressure cooker. Sections were treated with Proteinase K (Sigma), hybridized with 150 ng/ml riboprobe overnight at 60°, and incubated with anti-digoxigenin antibody conjugated with horseradish peroxidase (Roche Diagnostics 11207733910, 1:5000) and Tyramide-Cy3 amplification reagent (PerkinElmer). Sections were stained with ProLong antifade reagent with DAPI (Invitrogen).

Samples for whole-mount in situ hybridization were treated with Proteinase K (Sigma), neutralized with 2 mg/ml glycine, fixed in 4% paraformaldehyde with 0.1% glutaraldehyde (Electron Microscopy Sciences), treated with 0.1% sodium borohydride (Sigma), and hybridized with 0.5 μg/ml *Dkk4* riboprobe at 60° overnight. Embryos were subsequently treated with 2% blocking reagent (Roche Diagnostics) in maleic acid buffer with Tween-20 (MABT), incubated with an alkaline phosphatase-conjugated anti-digoxigenin antibody (Roche Diagnostics 112093274910, 1:5000) overnight at 4°, and developed 3–6 h in BM Purple (Sigma). Whole-mount preparations were photographed on a dissecting microscope under direct illumination, with oversaturation and vignetting artifacts corrected in processing. Nuclear fast red (Sigma) was used to stain 5 μM sections in Supplementary Fig. 3.

For analysis of *Dkk4* expression in fetal skin of $Ta^M/Ta^M$ and $Ta^b/Ta^b$ embryos (Fig. 2d), we quantified the difference between the two genotypes by counting the number of *Dkk4*-positive regions in 3.6 mm longitudinal segments of flank epidermis from sections of $Ta^M/Ta^M$ ($n = 4$ epidermal segments from three embryos) and $Ta^b/Ta^b$ ($n = 3$ epidermal segments from two embryos) embryos.

**Tissue preparation for single-cell RNA sequencing**. A single embryo at each of three developmental stages (Stage 15a, 15b, and 16a) was used to prepare a dissociated cell homogenate for single-cell RNA sequencing. Embryos were treated with Dispase (Stem Cell Technologies) and the epidermis was removed with gentle scraping using a No.15-blade scalpel. Epidermal sheets were dissociated with trypsin (Gibco) and gentle agitation. Red blood cells were removed with a red blood cell lysis solution (Miltenyi Biotec). Single-cell suspensions were incubated with a rat anti-CD49f antibody conjugated with phycoerythrin (ThermoFisher Scientific, clone NKI-GoH3 12-0495-82), followed by anti-phycoerythrin magnetic microbeads and enriched using a column-magnetic separation procedure (Miltenyi Biotec).

**Single-cell RNAseq**. We constructed and analyzed a total of nine scRNAseq libraries from fetal skin, of which three are described in detail here from stages 15a, 15b, and 16 (Supplementary Table 1). Each of the three libraries consisted of >4400 cells, with median gene and unique transcript counts ranging from 2,570 to 4,180 and 9,419 to 18,244, respectively, and 24,164−25,514 genes detected across all cell populations (Supplementary Table 4).

3′ single-cell barcoding and library preparation were performed using 10x Genomics single-cell RNA sequencing platform (10x Genomics). Libraries were constructed with version 2 (stage 16a) or version 3 (stage15 a/b) chemistry and sequenced on a single HiSeqX (Illumina) lane to generate 400–450 million paired-end 150 bp reads (Supplementary Table 4).

Barcoded reads were aligned to the domestic cat genome assembly (felCat9, NCBI Felis catus Annotation Release 104) using the 10x Genomics CellRanger

(v3.1) software. Annotations for 3′ untranslated regions were extended by 2 kb for all transcripts except when an extension overlapped a neighboring gene. The 'cellranger count' pipeline was used to output feature-barcode matrices, secondary analysis data, and browser visualization files. The 'cellranger reanalyze' function was used to exclude 247 cell cycle genes and *XIST* to minimize cell cycle and sex-associated variation in downstream analysis.

For the analysis of the stage 16a dataset, we used *k*-means clustering performed by the 'cellranger count' pipeline and gene markers to assign the identity of specific cell clusters (Supplementary Table 3 and Fig. 2a). Differential expression analysis between cell populations defined by clustering was measured using Cell Ranger's implementation of sSeq[47], which employs a negative binomial exact test. Genomic coordinates for the 604 genes differentially expressed at stage 16a were determined using the felCat9 genome assembly feature table file (www.ncbi.nlm.nih.gov/assembly/GCF_000181335.3/).

*k*-means clustering ($k = 10$) did not resolve subpopulations of basal keratinocytes at stage 15a, likely due to the relatively small number of *Dkk4*-positive cells (Fig. 2b, c). However, aggregation of expression matrices from stages 15a and 15b, using the 'cellranger aggr' function, resulted in co-alignment of stage15a ($n = 59$) and 15b ($n = 774$) *Dkk4*-positive cell populations in UMAP projections, and those projections are depicted in Fig. 2b.

For the analysis of the stages 15a and 15b datasets, we used an empiric threshold for *Dkk4* expression to distinguish between *Dkk4*-positive and *Dkk4*-negative cells, e.g., all cells in the data set demonstrating four-fold elevated *Dkk4* expression (relative to all cells, and equivalent to >2 *Dkk4* UMI counts per cell) were categorized as the *Dkk4*-positive population, and all basal keratinocytes (*Krt5*-positive; *Krtdap*-negative) with less than four-fold *Dkk4* levels (relative to all cells) were categorized as the *Dkk4*-negative population. We tested thresholds of two-, four-, eight-, and 16-fold, and observed little impact on the number of basal keratinocytes deemed *Dkk4*-positive and *Dkk4*-negative at each of the three stages (Supplementary Table 9). Differential expression analysis between the *Dkk4* populations was measured using Cell Ranger's implementation of sSeq.

For stage 15b, a subset of basal keratinocytes had a distinct gene expression signature characterized by the expression of *Engrailed* (*En1*) and *HoxC* genes, likely to reflect cells from the developing limb bud. The *En1/HoxC* signature was not observed at stages 15a or 16a, and therefore this population of cells was excluded from the stage15b differential expression analysis.

Supplementary Data 1 displays the expression levels, fold-change between *Dkk4*-positive and *Dkk4*-negative cells, and false discovery rates (FDR) for all genes. For analysis of differentially expressed genes across the different stages, we used two different approaches. Figure 3 depicts 508 genes differentially expressed (FDR < 0.05, expression level >0.25 transcripts/cell) for each of the three stages. However, using an FDR threshold can be biased by population size differences at each time point. For example, at stage 15a, when the ratio of *Dkk4*-positive to *Dkk4*-negative cells is low, comparison of 59 *Dkk4*-positive to 1569 *Dkk4*-negative cells yields 13 of 21761 genes with an FDR < 0.05, but a two-fold differential expression threshold yields 928 genes, of which 184 are in common with stages 15b and 16a (Fig. 3a, b). As an alternative and less stringent approach to differential expression analysis, we applied to each stage (1) a two-fold differential expression threshold and (2) a median normalized average of >0.1 transcript in one or more of the basal keratinocyte populations, which yields 928, 761, and 606 differentially expressed genes at stages 15a, 15b, and 16a, respectively (Supplementary Fig. 4).

For the 508 differentially expressed genes depicted in Fig. 3, hierarchical clustering by Euclidean distance using the pheatmap (v1.012) package in R v3.6.1–4.03 was used to identify three groups (cutree_rows = 3 option) of genes. Gene enrichment analysis using Enrichr[29] was used to identify a set of Kyoto Encyclopedia of Genes and Genomes (KEGG) pathways that are enriched for differentially expressed genes that are highly (cluster A) or moderately (cluster B) upregulated in *Dkk4*-positive keratinocytes. Enrichr uses two measures to evaluate enrichment, a Fisher exact test (*p*-value, Fig. 3b) and an expected rank test (odds ratio, Fig. 3b), for which null rankings are determined from a randomized set of gene lists[29]. Wnt pathway gene expression levels in keratinocytes (Fig. 3c) and expression levels for genes that define the different epithelial cell populations (Supplementary Fig. 2) were determined using the 10x Genomics Loupe cell browser (v3.1).

**Ticked genetic analysis.** Our initial evaluation of *Dkk4* as a candidate gene for *Ticked* was based on its genetic map location[19,37] and a survey of existing genome sequence data from the 99 Lives collection[20,38]. Whole-genome sequence data from 57 cats in the 99 Lives dataset were used for variant detection. Variant calling and annotation (NCBI Felis catus Annotation Release 103) were performed with Platypus[48] (v0.1.5) and SNPeff[49] (v4.3i), respectively, after alignment to the domestic cat genome (felCat8) with BWA[50] (v0.7.16). SNPsift (v4.3i) was used to identify protein-coding variants in *Vdac3*, *Polb*, *Dkk4*, and *Plat*.

Combined annotation-dependent depletion[51] (CADD, v1.4) was used to score the deleteriousness of coding variants within *Dkk4*, after converting to orthologous positions in the human genome (GRCh38/Hg38). SignalP-5.0[52] was used to predict signal peptide cleavage for the p.Ala18Val variant, and the PyMol browser (v2.3.3) was used to visualize and display the protein structure of the CRD1 domain of

Dkk4 (Fig. 5c), using the N-terminal region solution structure of recombinant human Dkk4 protein[39] (5O57, Protein Data Bank).

In addition to the 99 Lives collection, we collected additional DNA samples from cat shows and breeders, and carried out the association, haplotype, and segregation analysis to further evaluate the p.Ala18Val and p.Cys63Tyr variants in *Dkk4*. Beagle[53] v4.1 was used to infer haplotypes across a 15 Mbp interval from chrB1:30,000,000–45,000,000 (felCat8). Phased SNPs were thinned to 1 site/5 kb with VCFtools (v0.1.16) and converted to felCat9 assembly coordinates with the UCSC LiftOver tool. The color coding for reference (blue) and alternate (yellow) alleles (Supplementary Fig. 6a) reflect the felCat8 assembly, but the genomic coordinates presented are converted to the felCat9 assembly.

*Ticked* is required and/or strongly selected for in the Abyssinian, Burmese, and Singapura breeds. As shown in Table 1 and Supplementary Table 6, 37 Abyssinian and 26 Singapura cats carried at least one derivative *Dkk4* allele, the majority as homozygotes or compound heterozygotes. Among 13 Burmese cats, 11 were homozygous for the *Dkk4* p.Ala18Val variant, one was heterozygous, and one was homozygous for the ancestral allele (Table 1 and Supplementary Table 6). For other breeds including the Egyptian Mau, Ocicat, and Bengal cat, tabby markings are required; therefore *Ticked* should not be present due to its epistasis over *Tabby*[35]. In 31 such animals, none carried a derivative *Dkk4* allele (Table 1 and Supplementary Table 6). Finally, in feral cats and in some breeds such as the Oriental Longhair (OLH) and Oriental Shorthair (OSH), both Ticked and non-Ticked phenotypes are observed. In this group, we found that 29 of 29 Ticked cats were heterozygous or homozygous for the *Dkk4* p.Ala18Val variant, while no *Dkk4* deleterious variants were observed in 203 non-Ticked cats (Table 1 and Supplementary Table 6).

We evaluated Mendelian transmission and allelic interactions in an OSH pedigree (Supplementary Fig. 6b) and observed that the p.Ala18Val variant cosegregated perfectly with the Ticked phenotype (13 Ticked, 8 non-Ticked). In a Singapura pedigree that segregated both alleles (Supplementary Fig. 6c, the Ticked phenotype was observed in p.Ala18Val homozygotes ($n = 6$), p.Cys63Tyr homozygotes ($n = 1$), and compound heterozygotes ($n = 7$).

**Functional evaluation of *Dkk4* variants.** The domestic cat *Dkk4*, *Dkk4* p.Ala18Val, and *Dkk4* p.Cys63Tyr variants were synthesized as double-stranded DNA molecules to incorporate an optimized Kozak sequence and a sequence encoding a C-terminal, 6x Histidine-Flag epitope fusion on the 5′ and 3′ ends, respectively. The DNA fragments were cloned into NotI_XbaI sites of the mammalian expression vector, pTwist_CMV_BetaGlobin_WPRE_Neo (Twist Biosciences). Gene sequence and cloning junctions were confirmed by sequencing.

For transfections, 2 million HEK293 (ATCC CRL-1573) cells were seeded on a 10 cm dish and grown in DMEM + 10% FBS for 12–16 h. 10μg of *Dkk4* and *eGFP* (peGFP-N1, Takara Bio) expression plasmids were mixed and transfected using the calcium phosphate transfection method. After four hours, the transfection media was replaced with 10 ml serum-free CD293 media. After 48 h, the conditioned media was collected and concentrated to ~250 μl using a 10 kDa Amicon filter (Sigma), following the manufacturer's instructions. HEK293 cells were lysed in the dish with 1 mL of ice-cold RIPA buffer supplemented with a complete protease inhibitor tablet (Roche), and the soluble lysate was recovered after centrifugation.

Protein concentrations from concentrated conditioned media and soluble cell lysates were measured by Pierce BCA protein assay (Thermo Scientific) on a SpectraMax iD3 plate reader (Molecular Devices). Equivalent amounts were electrophoresed, blotted, and incubated with anti-GFP (abcam ab290, 1:1000) and anti-flag (Millipore Sigma, M2 clone F1804, 1:1000) antibodies followed by goat anti-rabbit horseradish peroxidase (Jackson ImmunoResearch 111-035-144, 1:5000) and goat anti-mouse horseradish peroxidase (Jackson ImmunoResearch 111-035-166, 1:5000), respectively, and ECL reagent (BioRad). Imaging and quantification of Western blots were performed using ChemiDoc Imaging System and the Image Lab Software Suite (v6.1, BioRad).

**Reporting summary**. Further information on research design is available in the Nature Research Reporting Summary linked to this article.

## Data availability
The raw and processed single-cell RNAseq data generated in this study have been deposited in the Gene Expression Omnibus database "GSE152946". GEO files include unfiltered feature-barcode matrices in HDF5 format, output by the CellRanger pipeline, and Illumina fastq files for stage 15a, 15b, and 16a single-cell RNAseq. The lists of differentially expressed genes and overlap with hair follicle placode genes are provided in Supplementary Data 1. All other relevant data supporting the key findings of this study are available within the article and its Supplementary Information files or from the corresponding author upon reasonable request. Source data are provided with this paper.

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

## Acknowledgements

The authors thank Hermogenes Manuel for technical assistance, Anthony Hutcherson and other members of The International Cat Association and Cat Fancy Association for assistance with breed sample collection, Trap-Neuter-Release programs in California for assistance with feral sample collection, Valerie Smith, Adriana Kajon, and Gulnaz Sharifzyanova for providing material from OSH, Singapura, and Savannah pedigrees, respectively, Helmi Flick and Jamila Agaeva for domestic and Savannah cat photographs, respectively. This research has been supported in part by the HudsonAlpha Institute for Biotechnology, and by a grant from the National Institutes of Health to G.S.B. (AR-067925).

## Author contributions

C.B.K., K.A.M., and G.S.B. conceived the project, designed experiments, and wrote the paper. C.B.K. and K.A.M. performed experiments and analyzed data; G.S.B. procured funding.

## Competing interests

The authors declare no competing interests.
