## [Peer Review File · Nature Communications]

Reviewers' Comments:

Reviewer #1:

Remarks to the Author:

In this manuscript Kaelin et al characterize both morphology and gene expression during skin development in cat embryos. They describe clear morphological differences in skin epidermis and then use single cell sequencing to identify genes that are differentially expressed in the thick and thin portions of the skin and in situ hybridization to validate their prime target, *Dkk4*. Furthermore, they show that the expression pattern of *Dkk4* mirrors the Tabby phenotype. Finally, they identify two coding variants in *Dkk4* and show they are associated with the Ticked phenotype in cats.

The work is stellar, the manuscript clearly written and the data logically presented and convincing. The figures clearly illustrate the main findings and I have a hard time finding faults with this manuscript. The results are important as they explain an important aspect of the genetics of patterning in animals.

Minor comment:

In Figure 3, I do not understand why the authors have indicated dotted lines from the numbers 121 and 90 in 3a to the table in 3c. I understand why they have chosen the 121 genes where you have overlap with all timepoints. However, I am not clear on why they have chosen 90 and not the 25 or 36 where there is also overlap in two datasets. This needs to be explained in the text or figure legend.

Reviewer #2:

Remarks to the Author:

The authors of this manuscript identified an early marker of skin color pattern formation in cats through a combination of histology, single cell sequencing and genetic mapping analyses. Through histology examination of carefully staged embryonic tissue, they found that thickening/thinning pattern in the epidermis in an early stage of embryogenesis correlates with color patterning in later stages. Through single cell transcriptomic analysis, they identified molecular markers for these early thickening epidermal regions, in particular, a Wnt signaling component *Dkk4*. Finally, through genetic mapping and sequencing of cat strains, they were able to identify that *Dkk* alleles account for the Ticked phenotypes. Overall, I support the publication of this article in Nature Communications with the following suggestions.

1. For Figure 2, it will be clearer to subset the keratinocytes population out for further analysis. Four clusters in Figure 2a and Figure 2b Stage 16a are seen and less clusters are seen in earlier stages. Why is the yellow population split into two far-apart clusters? Why is there a small blue population away from the main cluster? It would therefore be useful to plot markers of individual clusters (e.g. an expression heatmap for each cell clusters and top marker genes).
2. Moreover, it is not uncommon to use DEG fold change and transcripts number per cell to characterize cluster marker genes in single cell RNA data. A violin plot or feature plot of gene expression level is better for visualizing the marker genes such as *Dkk4* and *wif1*.
3. Again, the presentation of the DEG genes is a bit uncommon in Figure 3. The expression of the 121+63 genes across 6 cell populations (*Dkk4*+ vs *Dkk4*- , 3 stages) can be plotted as a heatmap. And a gene set enrichment analysis can be performed if possible and then the expression of wnt pathway components can be visualized as violin plots.
4. I find the statement of Line 128-130 is not directly supported from the gene expression signature analysis. Are there Wnt signaling reporters that can be visualized to support the statement?

Reviewer #3:

Remarks to the Author:

Kaelin et al analyze the molecular mechanism underlying color patterning in the cat. During cat embryo development, they identified alternating thick and thin regions of the epidermis, a novel

finding in itself. The pattern of this regions prefigures tabby pigmentation patterns in adult animals. Using single-cell RNAseq they found differential expression of DKK4 in the pre-patterned epidermis and hypothesized that expression of this secreted inhibitor of canonical wnt signaling may be responsible for morphological patterning of the thick epidermal region. Analyzing DNA samples in cat breeds, the authors then linked the Ticked phenotype to exonic variants of DKK4 (A18V and C63Y). The authors hypothesize that that these aa variants would negatively affect functional properties of DKK4 as wnt inhibitor and thereby cause Ticked phenotype. This is an attractive idea, but it lacks direct experimental support.

Overall, the study presents some nice developmental biology in an unusual vertebrate model system, benefitting from the availability of genetic pigmentation variants in the cat. However, to go from a GENETICS paper to Nature Communication, the authors would have to provide more experimental support of their model.

1. To support the suggestion of high/low wnt signaling in the thick/thin regions, markers for active Wnt signaling should be analyzed in embryo sections, e.g. Axin2, Sp5 by in situ hybridization, or nuclear b-catenin by antibody stain. Without direct data, Fig. 3 and especially panel "d" (about short- and long-range interactions) remains speculative and ill supported.

2. The claim that A18V and C63Y inactivate DKK4 function should be experimentally supported by carrying out WNT top flash reporter assays with transfected dkk4 wt and mutants (A18V, C63Y). This is a straightforward, easy experiment that can be done also in Corona times. This experiment is essential since e.g. in DKK1 experimental analysis of four naturally occurring missense mutations failed to influence DKK1 activity (<https://pubmed.ncbi.nlm.nih.gov/10965128/>), refuting a causal relationship between aa variants and phenotypes.

Minor points:

- a. It would be good if the quality of in situ hybridization can be improved (Fig.2e, 4e, high background).
- b. Legend Fig.2a,b needs to be improved. It is not described what is actually shown there. Most readers are not familiar with "UMAP visualization".

Reviewer #1 (Remarks to the Author):

In this manuscript Kaelin et al characterize both morphology and gene expression during skin development in cat embryos. They describe clear morphological differences in skin epidermis and then use single cell sequencing to identify genes that are differentially expressed in the thick and thin portions of the skin and in situ hybridization to validate their prime target, *Dkk4*. Furthermore, they show that the expression pattern of *Dkk4* mirrors the Tabby phenotype. Finally, they identify two coding variants in *Dkk4* and show they are associated with the Ticked phenotype in cats.

The work is stellar, the manuscript clearly written and the data logically presented and convincing. The figures clearly illustrate the main findings and I have a hard time finding faults with this manuscript. The results are important as they explain an important aspect of the genetics of patterning in animals.

Minor comment:

In Figure 3, I do not understand why the authors have indicated dotted lines from the numbers 121 and 90 in 3a to the table in 3c. I understand why they have chosen the 121 genes where you have overlap with all timepoints. However, I am not clear on why they have chosen 90 and not the 25 or 36 where there is also overlap in two datasets. This needs to be explained in the text or figure legend.

The choice of which subsets to highlight with dotted lines was somewhat arbitrary, and this feature of the figure has been removed in the revised manuscript. More important, as suggested by reviewer #2, the analysis and presentation of the scRNA-seq data has been revised. Differentially expressed genes are presented as a heat map, accompanied by a gene set enrichment analysis and violin plot of Wnt pathway genes, all in Figure 3. We think that overlapping subsets of differentially expressed genes as presented in the original Figure 3 does add value by providing a more direct and more sensitive (but less stringent) way to visualize the data, and what was formerly Figure 3 is now Supplementary Figure 4 (with the dotted lines removed). The actual data on differential gene expression has not changed, and is still presented in Supplementary Table 5.

Reviewer #2 (Remarks to the Author):

The authors of this manuscript identified an early marker of skin color pattern formation in cats through a combination of histology, single cell sequencing and genetic mapping analyses. Through histology examination of carefully staged embryonic tissue, they found that thickening/thinning pattern in the epidermis in an early stage of embryogenesis correlates with color patterning in later stages. Through single cell transcriptomic analysis, they identified molecular markers for these early thickening epidermal regions, in particular, a Wnt signaling component *Dkk4*. Finally, through genetic mapping and sequencing of cat strains, they were

able to identify that Dkk alleles account for the Ticked phenotypes. Overall, I support the publication of this article in Nature Communications with the following suggestions.

1. For Figure 2, it will be clearer to subset the keratinocytes population out for further analysis. Four clusters in Figure 2a and Figure 2b Stage 16a are seen and less clusters are seen in earlier stages. Why is the yellow population split into two far-apart clusters? Why is there a small blue population away from the main cluster? It would therefore be useful to plot markers of individual clusters (e.g. an expression heatmap for each cell clusters and top marker genes).

As suggested, additional analyses have been carried out on the stage 16a dataset by clustering the top upregulated genes in each UMAP group of epithelial cells and considering the non-epithelial cells as a single group (Supplementary Fig. 2). The distinct UMAP groups that cluster together at k=9 (yellow, Fig. 2a) is split into two populations of keratinocytes distinguished mainly by differential expression of Defb1, Lrp1, Krt5, Krt10, and Krt17 at k=10 (purple, yellow, Supplementary Fig. 2). The distinct UMAP group (blue, Fig. 2a) that clusters together with Dkk4-negative basal keratinocytes at k=9 is resolved into a third population of basal keratinocytes (red, Supplementary Fig. 2) characterized mainly by reduced expression of several Wnt pathway and cadherin genes (Supplementary Fig. 2c).

2. Moreover, it is not uncommon to use DEG fold change and transcripts number per cell to characterize cluster marker genes in single cell RNA data. A violin plot or feature plot of gene expression level is better for visualizing the marker genes such as Dkk4 and wif1.

3. Again, the presentation of the DEG genes is a bit uncommon in Figure 3. The expression of the 121+63 genes across 6 cell populations (Dkk4+ vs Dkk4- , 3 stages) can be plotted as a heatmap. And a gene set enrichment analysis can be performed if possible and then the expression of wnt pathway components can be visualized as violin plots.

As suggested, additional analyses of differentially expressed genes that distinguish Dkk4-positive and Dkk4-negative cells have been carried out and are presented as a heat map accompanied by a gene set enrichment analysis and violin plot of Wnt pathway genes (also described in the reply to reviewer #1). The new analysis is presented in what is now Figure 3 and what was formerly Figure 3 is now Supplementary Figure 4.

4. I find the statement of Line 128-130 is not directly supported from the gene expression signature analysis. Are there Wnt signaling reporters that can be visualized to support the statement?

New experiments have been carried out that demonstrate, in Dkk4-positive compared to Dkk4-negative cells, increased nuclear localization of Ctnnb1 and expression of Edar, a direct target of Wnt activation. This data is presented in a new Figure (Figure 4).

Reviewer #3 (Remarks to the Author):

Kaelin et al analyze the molecular mechanism underlying color patterning in the cat. During cat embryo development, they identified alternating thick and thin regions of the epidermis, a novel finding in itself. The pattern of these regions prefigures tabby pigmentation patterns in adult animals. Using single-cell RNAseq they found differential expression of DKK4 in the pre-patterned epidermis and hypothesized that expression of this secreted inhibitor of canonical Wnt signaling may be responsible for morphological patterning of the thick epidermal region. Analyzing DNA samples in cat breeds, the authors then linked the Ticked phenotype to exonic variants of DKK4 (A18V and C63Y). The authors hypothesize that these aa variants would negatively affect functional properties of DKK4 as Wnt inhibitor and thereby cause the Ticked phenotype. This is an attractive idea, but it lacks direct experimental support.

Overall, the study presents some nice developmental biology in an unusual vertebrate model system, benefitting from the availability of genetic pigmentation variants in the cat. However, to go from a GENETICS paper to Nature Communication, the authors would have to provide more experimental support of their model.

1. To support the suggestion of high/low Wnt signaling in the thick/thin regions, markers for active Wnt signaling should be analyzed in embryo sections, e.g. Axin2, Sp5 by in situ hybridization, or nuclear β -catenin by antibody stain. Without direct data, Fig. 3 and especially panel "d" (about short- and long-range interactions) remains speculative and ill supported.

As indicated in the reply to reviewer #2 (point 4), new experiments have been carried out that demonstrate, in Dkk4-positive compared to Dkk4-negative cells, increased nuclear localization of Ctnnb1 and expression of Edar, a direct target of Wnt activation. This data is presented in a new Figure (Figure 4). (Axin2 and Sp5 are expressed at very low levels in cat fetal skin (Supplementary Table 5) and could not be detected by in situ hybridization).

2. The claim that A18V and C63Y inactivate DKK4 function should be experimentally supported by carrying out WNT top flash reporter assays with transfected dkk4 wt and mutants (A18V, C63Y). This is a straightforward, easy experiment that can be done also in Corona times. This experiment is essential since e.g. in DKK1 experimental analysis of four naturally occurring missense mutations failed to influence DKK1 activity (<https://pubmed.ncbi.nlm.nih.gov/10965128/>), refuting a causal relationship between aa variants and phenotypes.

We agree that functional data on the A18V and C63V variants strengthens the claim of causality, although we have taken a different approach than suggested. As an aside, we note that in the 2000 reference on DKK1 as a candidate for human holoprosencephaly, the genetic evidence was relatively weak and the missense alterations were either conservative or failed to cosegregate fully with the phenotype, so the prior probabilities of causality are very different.

Our functional data on the A18V and C63V variants is based on trafficking and secretion—in brief, we engineered epitope-tagged normal and variant forms of cat Dkk4, expressed those in 293 cells, and measured the amount of protein produced and secreted after 48 hours. Our new

results show that the A18V or C63Y variants block or partially block secretion, respectively. The new data is presented in Figure 5d, and what was formerly Figure 5d, the effect of the A18V variant on pattern formation and Dkk4 expression in adult animals and fetal skin, respectively, is now Figure 6.

We also carried out top flash assays with Dkk4 variants, but our preliminary results and discussions with our colleague Dr. Roel Nusse have convinced us that the approach is not straightforward. Our expression data indicates that the most likely in vivo activator components are Wnt10b and Lrp4 (Figure 3, Supplementary Figure 4, Supplementary Table 5), but the top flash assays generally use Wnt1 or Wnt3a. Moreover, top flash assays will fail to capture variant effects altering Dkk4 diffusion in vivo, a potentially critical parameter given the anticipated role for Dkk4 as a reaction-diffusion component and precedent for variants affecting disulfide bonding to impair protein folding. Taking these factors into account, we believe that the production and secretion assays (now shown in Figure 5d) are a more direct and straightforward functional assay demonstrating Dkk4 loss of function.

Minor points:

a. It would be good if the quality of in situ hybridization can be improved (Fig.2e, 4e, high background).

The apparent high background in these images was a function of the lighting rather than the experimental conditions. Both sets of images have been improved. (What was Figure 4e is now Figure 6b).

b. Legend Fig.2a,b needs to be improved. It is not described what is actually shown there. Most readers are not familiar with “UMAP visualization”.

The legend has been reworded to improve clarity and accessibility.

Reviewers' Comments:

Reviewer #1:

Remarks to the Author:

The authors have addressed all my concerns. This is an elegant manuscript.

Reviewer #2:

Remarks to the Author:

The authors have satisfactorily addressed the questions I raised. And I support the publication of this manuscript.

Reviewer #4:

Remarks to the Author:

The authors have improved the ms and it may be published pending minor additions notably to legends, which continue to be an issue and the senior author should pay more attention to them:

1. Fig3d lacks any comment in legend
2. Fig. 5d Normalization of data to GFP and control Dkk4 needs to be properly described in legend.

Finally, regarding topflash assay suggestion, the authors argument of Wnt10b and Lrp4 is immaterial. Dkk4 function can be tested against Wnt1 or Wnt3a just fine in Hek293 cells.

Reviewer #1 (Remarks to the Author):

The authors have addressed all my concerns. This is an elegant manuscript.

Reviewer #2 (Remarks to the Author):

The authors have satisfactorily addressed the questions I raised. And I support the publication of this manuscript.

Reviewer #4 (Remarks to the Author):

The authors have improved the ms and it may be published pending minor additions notably to legends, which continue to be an issue and the senior author should pay more attention to them:

1. Fig3d lacks any comment in legend.

The following text has been added:

"A reaction-diffusion model for color pattern establishment in basal epidermis where Wnt pathway components participate in both short-range activation and long-range inhibition."

2. Fig. 5d Normalization of data to GFP and control Dkk4 needs to be properly described in legend.

The following text has been added:

Relative intensity on the y-axis refers to ratio of the Dkk4 band to GFP band intensity, normalized to non-mutant Dkk4; thus non-mutant Dkk4 relative intensity = 1.0 for both media and cell lysate.

Finally, regarding topflash assay suggestion, the authors argument of Wnt10b and Lrp4 is immaterial. Dkk4 function can be tested against Wnt1 or Wnt3a just fine in Hek293 cells.